# Multi-omics phenotyping of the gut-liver axis reveals metabolic perturbations from a low-dose pesticide mixture in rats

Robin Mesnage [1], Maxime Teixeira [2], Daniele Mandrioli [3], Laura Falcioni [3], Mariam Ibragim[1], Quinten Raymond Ducarmon[4], Romy Daniëlle Zwittink [4], Caroline Amiel[2], Jean-Michel Panoff[2], Emma Bourne[5], Emanuel Savage[5], Charles A. Mein[5], Fiorella Belpoggi[3] & Michael N. Antoniou [1]✉

Health effects of pesticides are not always accurately detected using the current battery of regulatory toxicity tests. We compared standard histopathology and serum biochemistry measures and multi-omics analyses in a subchronic toxicity test of a mixture of six pesticides frequently detected in foodstuffs (azoxystrobin, boscalid, chlorpyrifos, glyphosate, imidacloprid and thiabendazole) in Sprague-Dawley rats. Analysis of water and feed consumption, body weight, histopathology and serum biochemistry showed little effect. Contrastingly, serum and caecum metabolomics revealed that nicotinamide and tryptophan metabolism were affected, which suggested activation of an oxidative stress response. This was not reflected by gut microbial community composition changes evaluated by shotgun metagenomics. Transcriptomics of the liver showed that 257 genes had their expression changed. Gene functions affected included the regulation of response to steroid hormones and the activation of stress response pathways. Genome-wide DNA methylation analysis of the same liver samples showed that 4,255 CpG sites were differentially methylated. Overall, we demonstrated that in-depth molecular profiling in laboratory animals exposed to low concentrations of pesticides allows the detection of metabolic perturbations that would remain undetected by standard regulatory biochemical measures and which could thus improve the predictability of health risks from exposure to chemical pollutants.

[1] Gene Expression and Therapy Group, King's College London, Faculty of Life Sciences & Medicine, Department of Medical and Molecular Genetics, Guy's Hospital, London, UK. [2] UR Aliments Bioprocédés Toxicologie Environnements, EA 4651, University of Caen Normandy (UCN), Caen, France. [3] Ramazzini Institute (RI), Bologna, Italy. [4] Center for Microbiome Analyses and Therapeutics, Leiden University Medical Center, Leiden, The Netherlands. [5] Genome Centre, Barts and the London School of Medicine and Dentistry, Blizard Institute, London, UK. ✉email: michael.antoniou@kcl.ac.uk

Toxic effects of chemicals are not always accurately detected using the current battery of regulatory toxicity tests. Although Organization of Economic Co-operation and Development (OECD) testing guidelines are well established to measure acute toxic effects, they are not always sufficient to detect metabolic and endocrine disturbances, which can lead to chronic diseases, such as neurodegenerative disorders, diabetes, obesity and childhood cancers[1,2]. A delay between the introduction of new chemicals and the detection of their ill-effects in human populations has been a frequent occurrence stretching back many decades[3].

Human exposure to pesticides has been linked to the development of a large range of diseases caused by acute intoxication, repeated occupational exposure or residential proximity to pesticide use[4,5]. Furthermore, studies suggest that pesticides are major contributors to the development of a wide range of chronic diseases at environmental levels of exposure in human populations[6–12]. Effects of mixtures of pesticides at regulatory permitted levels are difficult, if not impossible, to predict by testing isolated compounds with current testing guidelines[13,14]. Although the assessment of the cumulative exposure of pesticides on the thyroid and nervous system is currently in progress[15], the evaluation of effects of mixtures is not being performed in government-led programmes. Among the different strategies to study chemical mixtures[16], some authors have proposed to estimate toxic effects by simulating real-life exposures in laboratory animals[17].

The diet is a major route of exposure to pesticide residues[15]. Dietary pesticide exposures mostly originate from the application of pesticides on crops during cultivation[18], but also from contamination of soil and water[19], as well as from post-harvest applications during storage[20]. With this in mind, we present here a study that tested the toxic effects resulting from subchronic exposure to a mixture of the six most frequently detected pesticide residues in EU foodstuffs[15,21] in Sprague–Dawley rats. This mixture consisted of azoxystrobin[22], boscalid[23], chlorpyrifos[24], glyphosate[25], imidacloprid[26] and thiabendazole[27], all combined at their respective regulatory acceptable daily intake (ADI) levels.

Glyphosate is the most frequently found herbicide residue in EU foodstuffs[21]. It is used in broad-spectrum herbicides and acts by inhibiting the plant enzyme 5-enolpyruvylshikimate-3-phosphate synthase[28]. Its residues are mostly found in cereals because glyphosate-based herbicides are frequently sprayed shortly before harvest to help crop desiccation and earlier harvest[29]. Imidacloprid and chlorpyrifos are key ingredients used in the two major categories of insecticides, neonicotinoids[30] and organophosphates[31], respectively. The pesticide mixture also included three fungicides: namely, azoxystrobin, a quinone outside inhibitor[32]; boscalid, a succinate dehydrogenase inhibitor[33]; and thiabenbazole, a mitotic spindle distorting agent[34].

The overall objective of our investigation was to identify metabolic perturbations caused by low-dose pesticide exposure to gain insight into mechanisms of toxicity, which could act as early biochemical markers of chronic ill-effects. High-throughput '-omics technologies' are increasingly used to evaluate the molecular composition of complex systems to understand the mode of action of chemicals and to provide metabolic signatures predictive of long-term health effects[35–37]. For instance, metabolomics has become established as a reliable method to identify biomarkers of numerous disease states[38–40], as well as for real-time diagnostics during surgery[41]. In contrast, best practice standards for the use of metabolomics in regulatory toxicology have only recently been proposed[42]. We employed a multi-omics strategy, including metabolomics, in an effort to identify metabolic perturbations, resulting from pesticide exposure. Our multi-omics approach combined shotgun metagenomics and metabolomics of caecum content, as well as serum metabolomics, and liver transcriptomics and DNA methylation profiles. We compared our multi-omics approach to standard clinical and biochemical measures recommended in OECD guidelines for the testing of chemicals, and followed by industry and government regulatory agencies.

Our findings show that unlike standard OECD blood biochemistry and organ histological analysis conducted for regulatory purposes, blood metabolomics, liver transcriptomics and genome-wide DNA methylation analysis highlighted the adaptation to metabolic stress induced by exposure to the mixture of pesticides. Our results suggest that the adoption of multi-omics as part of regulatory chemical risk assessment procedures will result in greater sensitivity, accuracy and predictability of outcomes from in vivo studies, with positive public health implications.

## Results

The aim of this study was to test the toxicity in vivo of a mixture of pesticides, the residues of which are among those most frequently found in the EU food chain. A group of 12 female Sprague–Dawley rats were administered with a combination of azoxystrobin, boscalid, chlorpyrifos, glyphosate, imidacloprid and thiabendazole for 90 days via drinking water to provide the EU ADI, and compared to an equivalent control group receiving plain drinking water (Fig. 1A, B). No differences were observed between the treatment and control groups of animals in terms of water consumption (Fig. 1C), feed consumption (Fig. 1D) and mean body weight (Fig. 1E) during the 90-day treatment period. Histological analysis showed there was a non-significant increase in the incidence of liver and kidney lesions (Fig. 2A, B). A serum biochemistry analysis only detected a small reduction in creatinine levels (Fig. 2C).

In order to obtain insight into possible chronic effects of this pesticide mixture, we used high-throughput molecular profiling techniques to search for changes, which could predict the development of disease. We first built orthogonal partial least squares discriminant analysis (OPLS-DA) models to assess the predictive ability of the different omics approaches used in this study. Serum metabolomics (pR2Y = 0.001, pQ2 = 0.001), liver transcriptomics (pR2Y = 0.04, pQ2 = 0.001) and to a lesser extent the caecum metabolomics (pR2Y = 0.16, pQ2 = 0.01), discriminated the pesticide-treated group from the concurrent control group (Table 1). Shotgun metagenomics in caecum and genome-wide methylation in liver did not discriminate between the two experimental groups.

Analysis of the host–gut microbiome axis using metabolomics revealed effects on the tryptophan–nicotinamide pathway (Table 2). A decrease in serum tryptophan levels, and its breakdown product indoleacetate, suggested that this amino acid is channelled to nicotinamide synthesis. The three metabolites nicotinamide N-oxide, 1-methylnicotinamide and nicotinamide all increased. In addition, the decrease in pyridoxal can also be linked to changes in nicotinamide metabolism (Table 2) as it is an important co-factor necessary for the synthesis of nicotinamide adenine dinucleotide.

Since the gut microbiome has known roles on nicotinamide metabolism, we analysed the composition and function of the caecum microbiome through shotgun metagenomics and metabolomics. The most discriminant metabolites between control and treatment groups of animals are glycerophospholipids, which accumulated in the caecum (Table 3). In addition, the increased levels of serotonin, which has both neurotransmitter and hormone functions and is synthesized from tryptophan, as well as the increased levels of pyridoxal and nicotinamide riboside, suggested that the effects of the pesticide mixture in the caecum

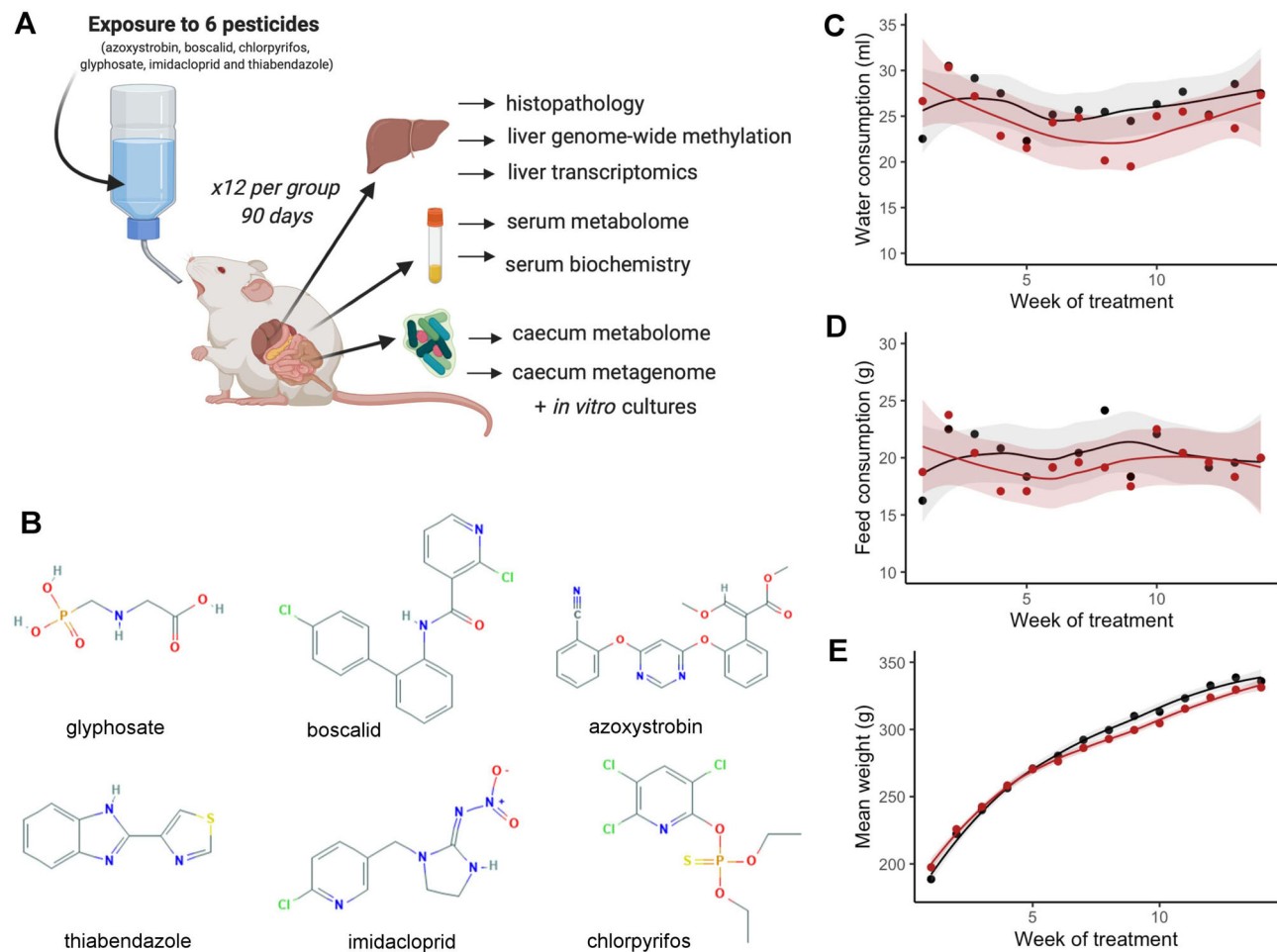

**Fig. 1 General toxicity assessment of a mixture of glyphosate, azoxystrobin, boscalid, chlorpyrifos, imidacloprid and thiabenzadole at their acceptable daily intake in Sprague–Dawley rats. A** Study design. Groups of 12 female rats were administered via drinking water with a mixture of six pesticides at the EU ADI for 90 days. Analyses following sacrifice are also shown (illustration created with BioRender.com). **B** Molecular structures of pesticide active ingredients tested. (Chemical structure from pubchem.com). Water consumption (**C**), feed consumption (**D**) and body weights with their 95% confidence interval band (**E**) remained unchanged (controls, black; pesticide-exposed, red).

microbiome are linked to the observed disruption in serum metabolite levels (Tables 2 and 3).

Shotgun metagenomics (Fig. 3A) showed that the caecum metagenome of the Sprague–Dawley rats was dominated by Firmicutes and Bacteroidetes. No species were detected in a negative extraction control, which was included to ensure that no bacterial contamination was introduced by laboratory reagents and procedures. Permutational ANOVA (PERMANOVA) analysis of the differences between Bray–Curtis distances calculated from the abundance determined with IGGsearch, did not show an effect of the pesticide treatment (Fig. 3C). No differences in species abundance were identified with a range of shotgun metagenomics taxonomic classifiers, such as IGGsearch, MetaPhlan2 or Kaiju (Supplementary Data 1). There was a large intragroup variability by comparison to the variability observed between the two groups of rats (species-level operational taxonomic units of IGGsearch shown in Fig. 3B), which prevented reaching definitive conclusions. This was also the case when we measured pathway abundances for tryptophan (Fig. 3D) and nicotinamide metabolism (Fig. 3E).

Since the low levels of pesticides that comprise the mixture studied here had limited effects on gut microbiome composition, we further evaluated if the mixture and the tested pesticide individually affected bacterial growth in vitro at a broad range of concentrations. We used two strains of *Escherichia coli* and two

strains of *Lactobacillus rhamnosus*, which are found in the human intestinal microbiota (Fig. 4). When the six pesticide active ingredients were tested individually, they did not affect the growth of the four bacterial strains (Figs. S1 and S2). However, when the bacterial strains were exposed to the mixture of all six pesticides, growth inhibition was detected in a strain-dependent manner. Although *L. rhamnosus* LB5 (Fig. 4A) and *E. coli* EC4 (Fig. 4C) had their growth inhibited by the pesticide mixture, *L. rhamnosus* LB6 (Fig. 4B) and *E. coli* EC2 (Fig. 4D) were not affected.

Mammals mostly produce the vitamin nicotinamide (a form of vitamin B3) from tryptophan in the liver before it is distributed to non-hepatic tissues. In order to understand if the changes in the gut and blood metabolome we observed are associated with disturbance in liver function, we compared transcriptome profiles in the liver of the two groups of rats by RNA sequencing (Supplementary Data 2). The results showed that the expression of 121 genes was increased and 117 genes had their expression decreased (Fig. 5A) by exposure to the pesticide mixture (adj-$p < 0.05$). A total of 96 Gene Ontology (GO) terms were enriched among the differentially expressed genes. Most of them were involved in the regulation of response to hormones (adj-$p = 0.0003$) and particularly to steroids (adj-$p = 0.0004$). The liver transcriptome also reflected the activation of stress response pathways (adj-$p = 0.02$). Interestingly, the two genes with the lowest $p$ values had their

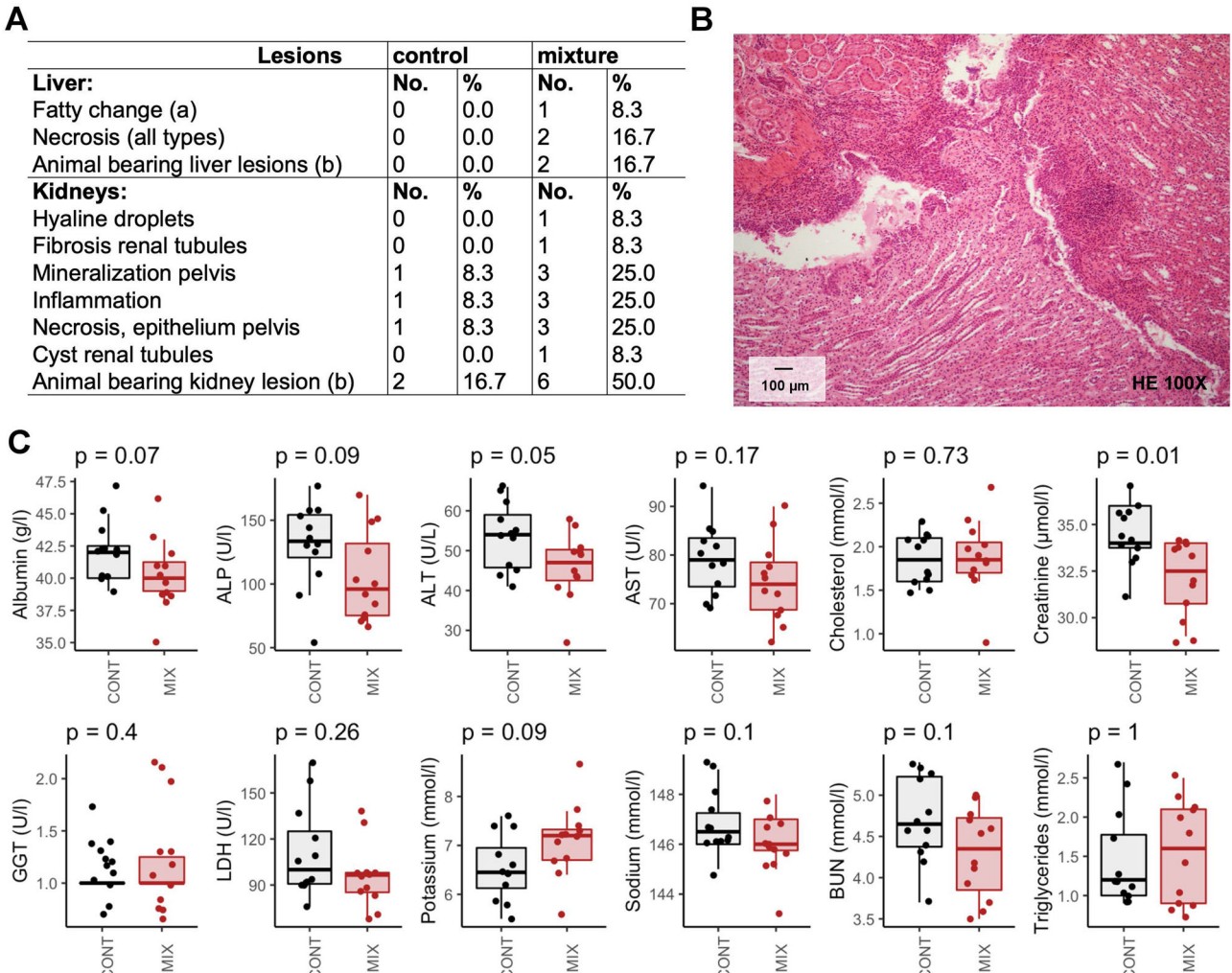

**Fig. 2 Analysis of regular clinical and biochemical markers provided limited insight into the effects of a mixture of pesticides at their acceptable daily intake in Sprague–Dawley rats.** Incidence in signs of anatomical pathologies in liver and kidneys (**A**). Focal inflammation of moderate to severe grade localized in the pelvic area in a rat exposed to the pesticide mixture; magnification ×100 (**B**). Standard serum biochemistry analysis showed only a decrease in creatinine levels (**C**).

**Table 1 Predictive ability of high-throughput omics approaches to evaluate the effects of a pesticide mixture in rats.**

| Omics technology | n | R2X | R2Y | Q2 | pR2Y | pQ2 |
|---|---|---|---|---|---|---|
| Serum metabolome | 749 | 0.34 | 0.99 | 0.80 | 0.001 | 0.001 |
| Caecum metabolome | 744 | 0.34 | 0.86 | 0.46 | 0.16 | 0.01 |
| Caecum metagenome | 2466 | 0.72 | 0.99 | 0.71 | 0.85 | 0.3 |
| Liver transcriptome | 18,170 | 0.48 | 0.97 | 0.49 | 0.04 | 0.001 |
| Liver genome-wide methylation | 136,555 | 0.06 | 0.88 | 0.00 | 0.4 | 1 |

OPLS-DA models were performed for each set of omics data. We show estimates of the total explained variation (R2X), variations between the different groups (R2Y) and the average prediction capability (Q2). We assessed the significance of our classification using permutation tests. The number of variables investigated (n) is also shown. New estimates of R2Y and Q2 values were calculated from this 1000 times permuted dataset (p values pR2Y and pQ2 for permuted R2Y and Q2, respectively).

expression increased (Fig. 5B) and coded the nuclear receptor subfamily 1 group D member 1 (*Nr1d1*, adj-$p = 3.2e-17$) and 2 (*Nr1d2*, adj-$p = 1.9e-18$), which play a role in carbohydrate and lipid metabolism, as well as in regulation of circadian rhythms. We finally compared our transcriptome findings to a list of gene expression signatures collected from various rat tissues after treatment with various drugs from the drugMatrix toxicogenomics database (Table S2). The alterations in the liver transcriptome we observed correlated well with those of livers from a study in which nicotinamide was administered to rats at a

dose of 750 mg/kg (adj-$p = 1.6e-7$). This correlates well with the changes in the biochemical composition of the serum–gut microbiome axis.

Reduced representation bisulfite sequencing (RRBS) of the liver samples was also performed to assess if alterations in epigenetic (DNA methylation) status may be responsible at least in part for the treatment-related changes in gene expression patterns. There were no differences in the percentage of methylated cytosines in CpG islands (42.0 ± 1.6% for controls vs 42.2 ± 2.2% for pesticide-exposed rats). We also analysed the distribution of the percentage

**Table 2 Serum metabolomics of host–gut microbiota interactions in rats exposed to a pesticide mixture reveals alterations in multiple metabolic pathways.**

| Compound | Pathway | FC | P | FDR | VIP |
|---|---|---|---|---|---|
| Pipecolate | Lysine metabolism | −6.5 | $3.5 \times 10^{-7}$ | 0.0002 | 2.79 |
| 3-Methylglutaconate | Leucine, isoleucine and valine metabolism | −4.1 | $1.1 \times 10^{-5}$ | 0.0027 | 2.64 |
| 4-Hydroxyphenylacetate | Phenylalanine metabolism | −4.1 | $1.7 \times 10^{-5}$ | 0.0028 | 2.47 |
| 1-Methylnicotinamide | Nicotinate and nicotinamide metabolism*** | 1.9 | $2.1 \times 10^{-4}$ | 0.0167 | 2.29 |
| Glutarate (C5-DC) | Fatty acid, dicarboxylate** | −3.3 | $2.4 \times 10^{-4}$ | 0.0167 | 2.29 |
| Nicotinamide N-oxide | Nicotinate and nicotinamide metabolism*** | 3.1 | $1.9 \times 10^{-4}$ | 0.0167 | 2.28 |
| 3-Hydroxyadipate* | Fatty acid, dicarboxylate** | −4.2 | $2.6 \times 10^{-4}$ | 0.0167 | 2.27 |
| Mevalonate | Mevalonate metabolism | −2.8 | $3.2 \times 10^{-4}$ | 0.0167 | 2.24 |
| Alpha-ketoglutarate | TCA cycle | −3.0 | $2.9 \times 10^{-4}$ | 0.0167 | 2.24 |
| N-Methyl-GABA | Glutamate metabolism | −2.5 | $6.5 \times 10^{-4}$ | 0.0281 | 2.17 |
| Eicosanedioate (C20-DC) | Fatty acid, dicarboxylate** | −3.2 | $6.0 \times 10^{-4}$ | 0.0281 | 2.13 |
| 3-Dehydrocholate | Secondary bile acid metabolism | 3.9 | $1.0 \times 10^{-3}$ | 0.0371 | 2.12 |
| Nicotinamide | Nicotinate and nicotinamide metabolism*** | 1.5 | $1.2 \times 10^{-3}$ | 0.0386 | 2.12 |
| Pyridoxal | Vitamin B6 metabolism | −3.2 | $1.4 \times 10^{-3}$ | 0.0390 | 2.11 |
| Indoleacetate | Tryptophan metabolism | −3.2 | $1.4 \times 10^{-3}$ | 0.0390 | 2.09 |
| Perfluorooctanesulfonate | Chemical | −2.5 | $9.8 \times 10^{-4}$ | 0.0371 | 2.07 |
| Taurine | Methionine, cysteine and taurine metabolism | 1.1 | $1.2 \times 10^{-3}$ | 0.0386 | 2.06 |
| 5-Methyluridine (ribothymidine) | Pyrimidine metabolism, uracil containing | −2.5 | $1.5 \times 10^{-3}$ | 0.0390 | 2.05 |
| Tryptophan | Tryptophan metabolism | −2.3 | $1.7 \times 10^{-3}$ | 0.0405 | 2.02 |
| Hexadecenedioate (C16:1-DC) | Fatty acid, dicarboxylate** | −4.2 | $2.5 \times 10^{-3}$ | 0.0524 | 2.01 |
| Methionine | Methionine, cysteine and taurine metabolism | −2.3 | $1.6 \times 10^{-3}$ | 0.0390 | 2.00 |

We presented fold changes (FC) for the metabolites that were found to have their variable importance in projection (VIP) scores > 2 in the OPLS-DA analyses. P values from a Welch's t test (P) are presented with the FDR. The statistical significance of a pathway enrichment analysis is also presented (*$p < 0.05$; **$p < 0.01$; ***$p < 0.001$).

**Table 3 Caecum metabolomics of host–gut microbiota interactions in rats exposed to a pesticide mixture reveals alterations in multiple metabolic pathways.**

| Compound | Pathway | FC | P | FDR | VIP |
|---|---|---|---|---|---|
| Serotonin | Tryptophan metabolism | 1.6 | 0.003 | 0.28 | 2.50 |
| Citrulline | Urea cycle; arginine and proline metabolism | 1.6 | 0.002 | 0.25 | 2.46 |
| Stearoyl sphingomyelin (d18:1/18:0) | Sphingomyelins** | 3.9 | 0.008 | 0.47 | 2.46 |
| Hexadecanedioate (C16-DC) | Fatty acid, dicarboxylate | 1.9 | 0.003 | 0.28 | 2.44 |
| 1-Palmitoyl-2-oleoyl-GPG (16:0/18:1) | Phosphatidylglycerol (PG)* | 2.2 | 0.001 | 0.25 | 2.43 |
| 1-Palmitoyl-2-oleoyl-GPE (16:0/18:1) | Phosphatidylethanolamine (PE)*** | 2.2 | 0.001 | 0.25 | 2.41 |
| 1-(1-Enyl-stearoyl)-2-arachidonoyl-GPE | Plasmalogen* | 3.5 | 0.01 | 0.47 | 2.40 |
| Nicotinamide riboside | Nicotinate and nicotinamide metabolism | 1.4 | 0.01 | 0.47 | 2.35 |
| Pantothenate | Pantothenate and CoA metabolism | 1.4 | 0.02 | 0.47 | 2.31 |
| 1,2-Dioleoyl-GPE (18:1/18:1) | Phosphatidylethanolamine (PE)*** | 2.8 | 0.0005 | 0.25 | 2.28 |
| 1-Palmitoyl-2-oleoyl-GPC (16:0/18:1) | Phosphatidylcholine (PC)* | 2.3 | 0.03 | 0.54 | 2.26 |
| Pyridoxal | Vitamin B6 metabolism | 1.3 | 0.007 | 0.47 | 2.24 |
| 1-(1-Enyl-palmitoyl)-2-arachidonoyl-GPE | Plasmalogen* | 2.5 | 0.04 | 0.55 | 2.16 |
| N-Acetylarginine | Urea cycle; arginine and proline metabolism | 1.3 | 0.02 | 0.50 | 2.16 |
| Palmitoyl sphingomyelin (d18:1/16:0) | Sphingomyelins** | 2.2 | 0.02 | 0.47 | 2.15 |
| 1-Palmitoyl-2-palmitoleoyl-GPC (16:0/16:1) | Phosphatidylcholine (PC)* | 2.2 | 0.04 | 0.55 | 2.15 |
| 1-Stearoyl-2-oleoyl-GPC (18:0/18:1) | Phosphatidylcholine (PC)* | 2.5 | 0.05 | 0.62 | 2.13 |
| Glycerophosphoglycerol | Glycerolipid metabolism | −3.7 | 0.04 | 0.55 | 2.12 |
| N-Carbamoylaspartate | Pyrimidine metabolism, orotate containing | 1.7 | 0.02 | 0.49 | 2.10 |
| Heptadecanedioate (C17-DC) | Fatty acid, dicarboxylate | 1.3 | 0.09 | 0.66 | 2.07 |
| 1-Stearoyl-2-arachidonoyl-GPE (18:0/20:4) | Phosphatidylethanolamine (PE)*** | 2.6 | 0.03 | 0.54 | 2.06 |
| Heptanoate (7:0) | Medium chain fatty acid | 1.6 | 0.008 | 0.47 | 2.06 |
| Palmitoyl dihydrosphingomyelin (d18:0/16:0) | Dihydrosphingomyelins* | 2.7 | 0.02 | 0.50 | 2.06 |
| Glutamate | Glutamate metabolism | 1.3 | 0.07 | 0.65 | 2.03 |
| 1-Stearoyl-2-oleoyl-GPE (18:0/18:1) | Phosphatidylethanolamine (PE)*** | 1.8 | 0.05 | 0.63 | 2.02 |

We presented fold changes (FC) for the metabolites that were found to have their variable importance in projection (VIP) scores > 2 in the OPLS-DA analyses. P values from a Welch's t test (P) are presented with the FDR. The statistical significance of a pathway enrichment analysis is also presented (*$p < 0.05$; **$p < 0.01$; ***$p < 0.001$).

methylation per base. Since a given base is generally either methylated or not in a given cell, it is expected that the distribution of percentage DNA methylation per base has a bimodal distribution, which is the case in our study (Fig. 6A). We also annotated the methylation calls. DNA methylation patterns in relation to gene transcription start sites (TSS) showed that CpG dinucleotides near TSS tend to be unmethylated, which confirmed that our RRBS analysis is of a good quality (Fig. 6B). We

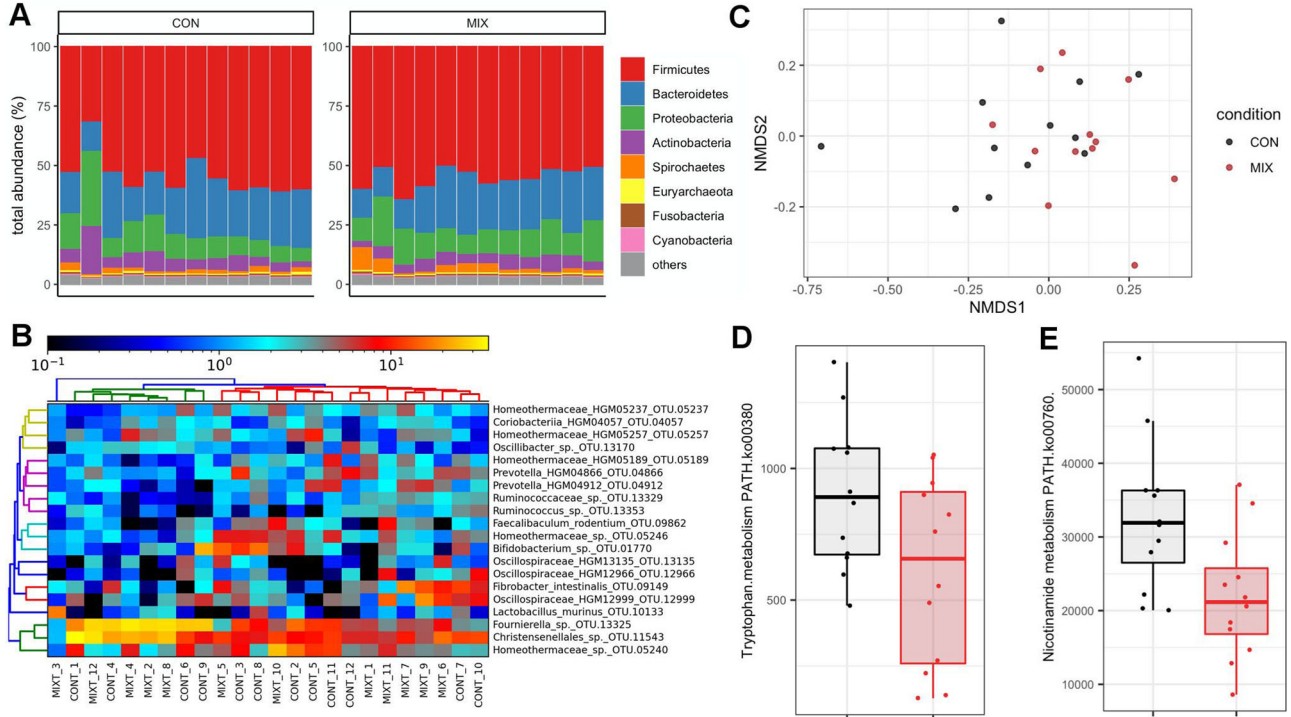

**Fig. 3 Shotgun metagenomics shows no alterations in the caecum microbiota composition upon exposure to the mixture of six pesticides. A** Gut caecum microbiota composition profiles at the phylum level. **B** Classification of samples from the most frequently found species-level operational taxonomic units with IGGsearch failed to show any alterations in different bacterial populations in response to the pesticide mixture. **C** Principal coordinates analysis plot using the NMDS ordination of Bray–Curtis distances. Pathway analysis shows reduction in tryptophan (**D**) and nicotinamide (**E**) metabolism potential.

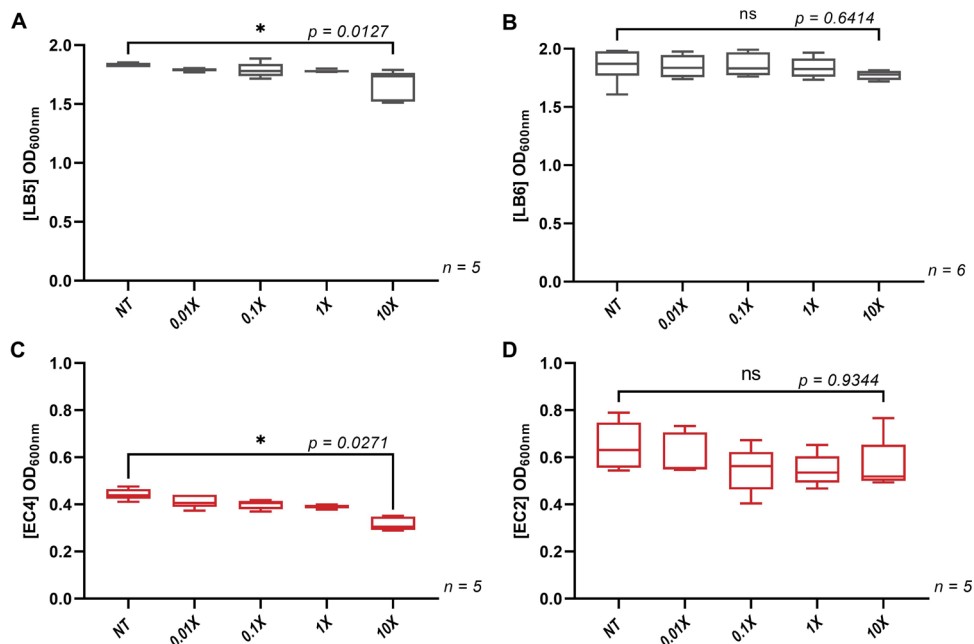

**Fig. 4 Effects of the pesticide mixture on *L. rhamnosus* and *E. coli* in vitro.** Bacterial growth of the strain *L. rhamnosus* (LB5) (**A**) is inhibited after an exposure at ten times the ADI (10×) for the mixture, while another strain of *L. rhamnosus* (LB6) (**B**) was not inhibited at the same dose. Similarly, *E. coli* (EC4) growth (**C**) was inhibited at lower doses in comparison with the other *E. coli* strain (EC2) (**D**).

identified 4255 differentially methylated CpG sites (FDR < 0.05) with a modest methylation difference (>10%) between the group of rats exposed to the pesticide mixture and the control animals (Fig. 6C, D and Supplementary Data 3). They were mostly located in intergenic regions (50.1%) and introns (37.0%), and to a lesser extent in exons (6.2%) and promoters (6.6%), at an average distance of 56 kb from TSS. Only 114 CpG sites presented differences in methylation levels over 25% (max 31.6%). A total of 24 differentially methylated CpG sites were present in promoters (Table 4). Interestingly, the lowest *p* value was for

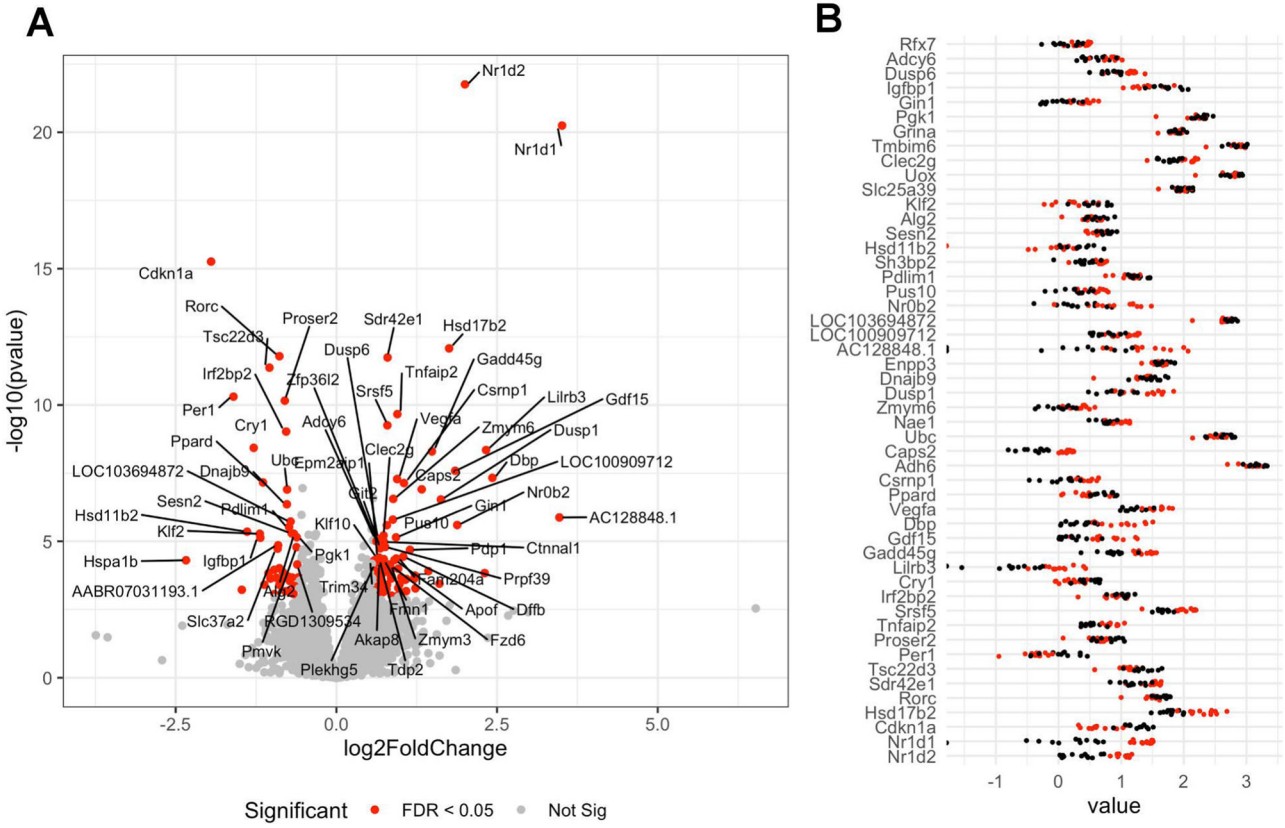

**Fig. 5 Liver transcriptomics of Sprague–Dawley rats exposed for 90 days to the mixture of six pesticides.** Genes were considered as differentially expressed if their count were found to be statistically significant after an analysis with DESeq2. **A** A volcano plot showing the fold changes and statistical significance in the expression of genes affected by exposure to the pesticide mixture. **B** The effect size for the 50 most affected transcripts. Log10-normalized abundances from the DESeq2 analysis were used to facilitate the visualization of differences (red dots, pesticide treated; black dots, untreated controls).

hypermethylation of the promoter of the gene coding coenzyme Q10A, a protein attenuating high-fat diet-induced non-alcoholic fatty liver disease[43]. We also evaluated the correlation between gene expression and DNA methylation. There was no correlation between the fold changes in gene expression and percentage methylation changes, and it was thus not possible to attribute changes in gene expression caused by the mixture to change in DNA methylation at their promoters (Fig. S3). However, as gene regulatory elements can be present within introns and at distantly located enhancers, changes in DNA methylation status in non-promoter regions as we have found can also in principle impact their function and alter levels of expression.

Our various omics analyses suggested oxidative stress resulting from exposure to the pesticide mixture. We therefore also assessed oxidative damage to liver DNA as measured by the rate of apurinic/apyrimidinic (A/P) site formation. Our results show that A/P site formation was unchanged between control and test groups of animals (Fig. S4).

Overall, exposure to the mixture of pesticides tested altered the gut–liver axis and caused changes in metabolites from the tryptophan–nicotinamide conversion pathway, which reflects a metabolic adaptation to oxidative stress.

## Discussion

We report here the first direct comparison of standard histopathology and serum biochemistry measures and multi-omics analysis of multiple physiological compartments of rats exposed to a mixture of six pesticides (azoxystrobin, boscalid, chlorpyrifos, glyphosate, imidacloprid and thiabendazole) that are most

frequently found in EU foodstuffs. Each pesticide was administered at its regulatory permitted EU ADI, and thus the expectation was that no effects or signs of toxicity would be observed. However, our results show that the low-dose mixture of pesticides we tested caused metabolic alterations in the caecum and blood metabolome, with consequences on liver function, which was mostly reflected by changes in the conversion of tryptophan to nicotinamide. Notably, these metabolic alterations were not detected by regular clinical and biochemical analyses as currently recommended in OECD guidelines and required by government regulatory agencies, but with the new generation of high-throughput 'omics' methodologies.

Unlike standard blood biochemical and organ histological analysis, an in-depth molecular profiling using a combination of high-throughput '-omics' methods revealed metabolic effects of the mixture of six pesticides. Histological analysis showed a non-significant increase in liver and kidney lesions (Fig. 2A, B). Considering the relatively short duration of exposure (90 days), and the limited number of animals per test group, we cannot rule out the possibility that these non-statistically significant increases were attributable to the treatment. Animal bioassays are generally extended to 12 months and performed on a larger number of animals (OECD Test Guideline 452: Chronic Toxicity Studies) to detect chronic toxicity from exposure to a given chemical. More studies are needed to determine if a longer treatment period with the pesticide mixture leads to adverse health effects.

Our results show that the inclusion of 'omics' high-throughput approaches in the battery of tests used to study the effects of chemicals promises to substantially heighten their sensitivity and

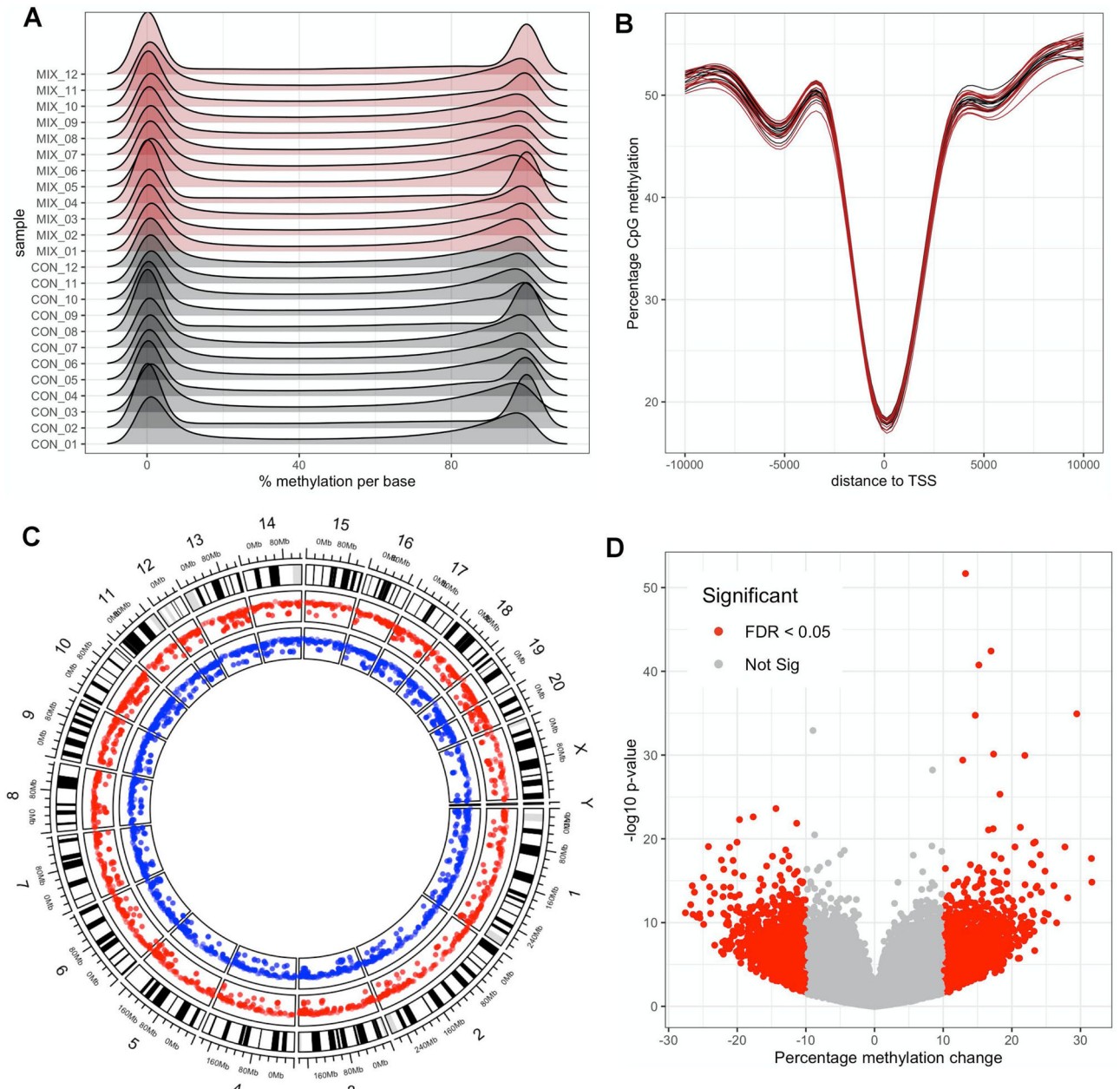

**Fig. 6 Reduced representation bisulfite sequencing of liver samples from Sprague–Dawley rats exposed for 90 days to the mixture of six pesticides.**
**A** Percentage DNA methylation profile shows a bimodal distribution of methylation calls for each sample. **B** CpG methylation decreases around transcription start sites. **C** Circos plot shows that differentially methylated CpG sites (blue track, hypomethylated; red track, hypermethylated) are scattered around the genome. **D** Volcano plots of differentially methylated CpG sites shows that a large number of CpG loci are differentially methylated across the rat liver genome with moderate methylation changes.

accuracy. This in turn will enhance the ability of such tests to predict risks of toxicity for regulatory decision-making purposes[44,45]. This could ultimately reduce the duration of animal bioassays and the number of animals needed, which would be in accord with ongoing efforts to improve animal welfare in research and testing. Omics methods first found a place in the field of toxicology when the US National Research Council published a vision and strategy statement on toxicity testing in 2007 calling for a transition toward high-throughput predictive and mechanistic chemical toxicity assessment[46]. This was mostly reflected by the development of adverse outcome pathways (AOPs), which are sequences of molecular and cellular events known to reflect the development of a pathological process[47]. Some omics technologies such as transcriptomics can be used to

predict key events and outcomes in networks of AOPs[48]. Contemporary toxicology is increasingly using artificial intelligence models to predict toxicological properties[49]. These models can be based on pathway perturbations identified using omics data as suggested by a recent study, showing that the training of a machine learning algorithm on the transcriptome changes caused by different endocrine disruptors in fish ovaries, adequately identified anti-androgenic properties of the herbicide linuron[50].

Although caecum metabolomics showed that there was an effect on gut microbial pathways involved in tryptophan and serotonin metabolism (Table 3), shotgun metagenomics failed to show an effect on gut microbiome functional potential (Fig. 3). Gut microbiome metagenomics and metabolomics are part of an emerging field of research and results from studies can be

**Table 4 Differentially methylated CpG sites located at gene promoters.**

| Chr | Coordinates | Strand | P | FDR | % Diff | Gene name |
|-----|-------------|--------|---|-----|--------|-----------|
| Chr7 | 2,787,189 | − | 5E−57 | 5E−53 | 15.4 | Coenzyme Q10A |
| Chr3 | 57,717,495 | + | 8E−55 | 5E−51 | −11.6 | Cytochrome b reductase 1 |
| Chr14 | 4,250,209 | + | 8E−50 | 1E−46 | 13.5 | Uncharacterized LOC102551276 |
| Chr7 | 138,705,521 | + | 4E−39 | 2E−36 | 11.1 | PC-esterase domain containing 1B |
| Chr1 | 219,852,329 | − | 6E−33 | 2E−30 | 10.8 | Leucine-rich repeat and fibronectin type III domain containing 4 |
| ChrX | 134,538,342 | + | 3E−32 | 7E−30 | −11.4 | Similar to CXXC finger 5 |
| Chr5 | 159,427,478 | + | 2E−28 | 3E−26 | −10.1 | Peptidyl arginine deiminase 2 |
| Chr15 | 33,120,272 | + | 4E−27 | 7E−25 | −11.6 | RRAD and GEM like GTPase 2 |
| Chr3 | 147,818,650 | − | 3E−25 | 5E−23 | −10.3 | Tribbles pseudokinase 3 |
| Chr17 | 80,793,952 | + | 1E−24 | 2E−22 | 19.2 | Uncharacterized LOC102550536 |
| Chr20 | 3,350,268 | + | 3E−23 | 4E−21 | 16.9 | Alpha tubulin acetyltransferase 1 |
| Chr19 | 40,616,367 | + | 2E−21 | 2E−19 | −11.7 | Uncharacterized LOC103694327 |
| Chr20 | 4,362,151 | + | 1E−19 | 1E−17 | 12 | Advanced glycosylation end product-specific receptor |
| Chr13 | 53,571,396 | − | 4E−19 | 4E−17 | 13.6 | Uncharacterized LOC102547588 |
| Chr1 | 87,044,466 | + | 5E−17 | 3E−15 | 16.8 | Galectin 7 |
| Chr8 | 70,015,572 | − | 1E−16 | 9E−15 | 13.3 | Uncharacterized LOC102548470 |
| Chr2 | 53,310,072 | − | 5E−16 | 3E−14 | 10.8 | Growth hormone receptor |
| Chr6 | 28,234,694 | + | 4E−15 | 2E−13 | 11 | DNA methyltransferase 3 alpha |
| Chr5 | 154,524,190 | + | 1E−11 | 4E−10 | −11.7 | E2F transcription factor 2 |
| Chr14 | 45,398,591 | − | 1E−11 | 3E−10 | −11.1 | Uncharacterized LOC102552762 |
| Chr10 | 107,157,939 | + | 2E−11 | 6E−10 | −10.7 | Uncharacterized LOC103693482 |
| Chr1 | 220,883,570 | − | 2E−09 | 3E−08 | −13.3 | Melanoma-associated antigen G1 like |
| Chr5 | 172,527,225 | − | 9E−08 | 1E−06 | 10.3 | Uncharacterized LOC108351067 |
| Chr9 | 4,547,982 | + | 9E−06 | 6E−05 | 10.7 | Uncharacterized LOC108351871 |

Reduced representation bisulfite sequencing was performed on liver samples. Differentially methylated CpG sites present in promoters were filtered and their variations summarized.

confounded by a large number of factors[51]. This could be amplified in our case by the housing of three rats per cage, since rats are coprophagous and thus exchange their gut microbiota[52]. It is also not clear if the gut microbiome is mediating the effects of this pesticide mixture, or if the change in liver metabolism is influencing gut microbiome composition and function. The gut microbiome and the liver evidently influence each other, and a large number of studies have shown a bidirectional communication involving bile acids, antimicrobial molecules or dietary metabolites[53]. It is also possible that the pesticide mixture tested here had an effect on bacterial metabolism, which did not affect growth properties. This possibility is supported by a recent study of the soil filamentous fungus *Aspergillus nidulans*[54]. This study found that exposure to the glyphosate-based herbicide Roundup GT+ caused alterations in secondary metabolism at a concentration that, however, caused no change in growth.

Changes in caecum and serum metabolome and liver transcriptome profiles were very consistent, and suggested that the mixture of pesticides tested in this study triggered an oxidative stress response. Low levels of pyridoxal, a form of vitamin B6, are known to be associated with changes in tryptophan levels and to cause inflammation[55,56]. This also explains the increase in nicotinamide levels, which is considered to be a marker for the generation of ROS[57]. Nicotinamide is a form of vitamin B3, which can act as a stress signal mediating compound, released when DNA-strand breakage is caused by oxidative damage[58]. Nicotinamide has been shown to protect from hepatic steatosis by increasing redox potential[59]. An improvement in hepatic transaminase concentration is detectable when hypertriglyceridemic patients are treated with nicotinamide[60]. Overall, this suggests that the increase in nicotinamide levels we observe reflects a metabolic adaptation to oxidative stress induced by exposure to the mixture of pesticides.

The changes in serum metabolite levels we see in this study do not reflect a diseased state, but probably metabolic adaptation, which can lead to the development of a pathological state if the damage produced exceeds the capacity for repair. The conversion

of tryptophan to nicotinamide is known to decrease in rats presenting a steatotic liver[59], which is the opposite to what we observed. Similarly, pipecolate levels are known to be increased in patients with liver disease, with the level of increase being proportional to the severity of liver damage[61]. In our study, pipecolate serum levels displayed the opposite trend; that is, they were decreased in animals exposed to the pesticide mixture (Table 2). This could reflect the establishment of a state of hormesis, a phenomenon by which mild-induced stress can give rise to a positive physiological counter-response inducing maintenance and repair systems[62]. This is well described for the effect of pesticides for both target[63] and non-target[64] species. Other known hormetic stressors include exercise[65] and fasting[66,67]. Although physiology may initially improve by initiating protective measures in the face of exposure to mild stressors, this can ultimately give rise to a pathological status if the intensity of the stimulation exceeds cellular capacity for homoeostasis[68]. A hormetic response may explain why changes in redox status did not result in increased DNA damage (Fig. S4). If damage was caused to DNA, it would have been repaired.

A dose–response study to identify the point of departure for these effects could inform on whether metabolic perturbations exceed cellular capacity for homoeostasis. In addition, because the liver is a sex-dimorphic organ with sex differences in liver metabolism and in the sensitivities to liver diseases[69], the results of this study in females cannot be extrapolated to males. This is supported by the results of another study which revealed the sexually dimorphic obesogenic and diabetogenic effects of a low-dose pesticide mixture using a combination of omics methods[14]. Although rats exposed to the pesticide mixture in this study presented more signs of pathology than animals in the control group, a longer period of exposure would be needed to determine if liver and/or kidney function will ultimately be impaired. In support of this possibility are findings from other studies, which showed changes in oxidative stress markers or inflammation profiles after exposure to low doses of pesticides with different profiles between 6, 12 and 18 months of administration[70,71].

In conclusion, our study reveals that metabolic effects following exposure to a pesticide mixture, which can be typically found in EU foodstuffs, and administered at the ADI, can be detected using caecum and serum metabolomics, and liver transcriptomics and DNA methylation profiling. Crucially, these molecular biological and metabolic changes would not be detected using conventional biochemical and histopathological investigations, which regulators currently rely upon for chemical risk assessment. Thus our results highlight the advantages of incorporating high-throughput '-omics' methods into OECD Guidelines for the Testing of Chemicals[72]. Although additional studies are needed to determine if longer exposure to the pesticide mixture we tested leads to adverse effects, our results demonstrate that high-throughput 'omics' analyses as applied herein can reveal molecular perturbations, which can potentially act as more sensitive and accurate predictors of long-term health risks arising from pesticide exposures. This in turn can lead to more appropriate regulatory public health protection measures.

## Methods

**Experimental animals**. The experiment was conducted according to Italian law regulating the use and humane treatment of animals for scientific purposes (Decreto legislativo N. 26, 2014. Attuazione della direttiva n. 2010/63/UE in materia di protezione degli animali utilizzati a fini scientifici.—G.U. Serie Generale, n. 61 del 14 Marzo 2014). Before starting the experiment, the protocol was approved and formally authorized by the ad hoc commission of the Italian Ministry of Health (authorization N. 447/2018-PR). The experiment was conducted on young adult female Sprague–Dawley rats (8 weeks old at the start of treatment).

**Animal management**. Female Sprague–Dawley rats from the Cesare Maltoni Cancer Research Center breeding facility were used. Female animals were chosen in order to make the results of this investigation comparable to our previous studies[73–75]. The animals were generated in-house following an outbreed plan and were classified as conventional (minimal disease) status. All the experimental animals were identified by ear punch according to the Jackson Laboratory system. After weaning, and before the start of the experiment, animals were randomized in order to have at most one sister per litter of each group; homogeneous body weight within the different groups was ensured. Animals of 6 weeks of age were acclimatized for 2 weeks before the start of the experiment.

Rats were housed in polycarbonate cages (41 × 25 × 18 cm) with stainless wire tops and a shallow layer of white wood shavings as bedding. The animals were housed in the same room, three per cage, maintained at the temperature of 22 ± 3 °C and relative humidity of 50 ± 20%. Lighting was provided by artificial sources and a 12-h light/dark cycle was maintained. No deviations from the above-mentioned values were registered. Cages were identified by a card indicating study protocol code, experimental and pedigree numbers, and dosage group. The cages were periodically rotated on their racks to minimize effects of cage positions on animals.

**Diet and treatments**. Experimental groups consisted of 12 female Sprague–Dawley rats of 8 weeks of age, treated for 90 days. Female animals were chosen to make the results of this investigation comparable to our previous studies, showing that the long-term exposure to a glyphosate-based herbicide was associated with the development of liver disease[73,74]. Animals received ad libitum a rodent diet supplied by SAFE (Augy, France). The feed was analysed to identify possible contaminants or impurities and these were found to be below levels of detection for all substances tested (Supplementary Data 4). The treatment group of animals was administered daily with a mixture of glyphosate (0.5 mg/kg bw/day)[25], azoxystrobin (0.2 mg/kg bw/day)[22], boscalid (0.04 mg/kg bw/day)[23], chlorpyrifos (0.001 mg/kg bw/day)[24], imidacloprid (0.06 mg/kg bw/day)[26] and thiabendazole (0.1 mg/kg bw/day)[27], via drinking water. The concentration of pesticides in tap water to give a dose equivalent to the ADI was calculated weekly on the basis of mean body weight and mean daily water consumption. Tap water from the local mains water supplier was administered, alone or with the test compounds, to animals in glass bottles ad libitum. Every 24 h, drinking water was discarded and the bottles cleaned and refilled. The presence of pesticides was not measured in the tap water. The regulation in the EU has maximum residue limits for a given pesticide in tap water of 0.1 µg/l, and up to a maximum of 0.5 µg/l for all pesticides that may be present. Given these low legal tolerance limits, it was not deemed necessary to analyse tap water for pesticide contamination, since if present would be extremely low and unlikely to have any physiological consequences on the experimental animals. Glyphosate, azoxystrobin, boscalid, chlorpyrifos, imidacloprid and thiabendazole were purchased from Merck KGaA (Sigma Aldrich®, Germany) with a purity ≥95%.

Although the study was performed on 12 animals per group, it was originally conceived to analyse 10 animals per group with 2 animals used as a contingency in case of unexpected death, as recommended by OECD guidelines for the testing of chemical toxicity. All animals survived and 12 animals per group were analysed for clinical biochemistry, histopathology, transcriptomics, methylation profiling and shotgun metagenomics, while ten animals per group were randomly chosen for the metabolomics analyses.

**Clinical observations**. Animals were checked for general status three times a day, 7 days a week, except non-working days when they were checked twice. Status, behaviour and clinical parameters of experimental animals were determined weekly beginning 2 weeks prior to commencement of treatments until the end of the experiment (at 90 days). Before final sacrifice and after ~16 h in a metabolic cage, water consumption was registered for each animal. Body weight, water and food consumption of experimental animals were measured before the start of the treatment and then weekly for 90 days. All the experimental animals were weighed just before sacrifice.

**Histopathology evaluation**. Each animal was anesthetized by inhalation of a mixture of 70% $CO_2$ and 30% $O_2$ and sacrificed by exsanguination from the vena cava. All sacrificed animals were subjected to complete necropsy. The gross necropsy was performed by initial physical examination of external surfaces and orifices followed by an internal in situ examination of tissues and organs. The examination included cranial cavity and external surfaces of the brain and spinal cord, thoracic abdominal and pelvic cavities with their associated organs and tissues, and muscular/skeletal carcass.

Liver and kidneys were alcohol-fixed, trimmed, processed and embedded in paraffin wax. Sections of 3–6 µm were cut for each specimen of liver and kidneys, and stained with haematoxylin and eosin. All slides were evaluated by a pathologist and all lesions of interest were reviewed by a senior pathologist.

The histopathological nomenclature of lesions adopted was in accord with international criteria; in particular non-neoplastic lesions were classified according to the international nomenclature INHAND (International Harmonization of Nomenclature and Diagnostic Criteria) and RITA (Registry of Industrial Toxicology Animal Data). Incidence of non-neoplastic lesions was evaluated with a Fisher's exact test (one- and two-tailed; one-sided results were also considered, since it is well established that only an increase in incidence can be expected from the exposure, and incidences in the control group are almost always 0).

**Blood biochemical analysis**. At the time of sacrifice, ~7.5 ml of blood was collected from the vena cava. The blood collected from each animal was centrifuged in order to obtain serum, which was aliquoted into labelled cryovials and stored at −70 °C. Serum biochemistry was performed under contract by IDEXX BioAnalytics (Stuttgart, Germany), an ISO 17025 accredited laboratory. Briefly, sodium and potassium levels were measured by indirect potentiometry. Albumin was measured by a photometric bromocresol green test. ALP was measured by IFCC with AMP-buffer method, cholesterol by Enzymatic colour test (CHOD-PAP), blood urea nitrogen by enzymatic UV-Test, gamma-glutamyl-transferase by Kinetic colour test International Federation of Clinical Chemistry (IFCC), aspartate and alanine aminotransferase by kinetic UV-test (IFCC + pyridoxal-5-phosphate), creatinine by kinetic colour test (Jaffe's method), lactate dehydrogenase by IFCC method and triglycerides using an enzymatic colour test (GPO-PAP) on a Beckman Coulter AU 480.

**DNA and RNA extraction**. DNA and RNA were extracted from rat liver tissue excised at the time of sacrifice and which had been stored at −70 °C, using the All-Prep DNA/RNA/miRNA Universal Kit (Qiagen, Hilden, Germany), using the manufacturer's instructions for 'Simultaneous purification of genomic DNA and total RNA from animal and human tissues' with no alterations. Tissue weight used was ≤30 mg, and samples were eluted in 30 µl RNase-free water. RNA samples were quantified with the Nanodrop 8000 spectrophotometer V2.0 (Thermo Scientific, USA) and integrity was checked using the Agilent 2100 Bioanalyser (Agilent Technologies, Waldbronn, Germany). All samples had RNA integrity numbers (RIN) ≥ 7. DNA quantity was measured using the Qubit® 2.0 Fluorometer (Life Technologies) with the dsDNA broad range reagent, followed by quality assessment using the Agilent 2200 Tapestation and Genomic DNA screentape (Agilent Technologies). Samples displayed high molecular weight with DINs (DNA integrity score) ranging from 7.5 to 10, and average concentrations from 370 ng/µl. All samples showed a majority of high molecular weight material and all were taken forward for processing.

**Transcriptomics analysis**. For RNA-seq transcriptomics, 100 ng of total RNA was used for each sample for library preparation. The mRNA libraries were prepared using the NEBNext® Poly(A) mRNA Magnetic Isolation Module in combination with the NEBNext® Ultra™ II Directional RNA Library Prep Kit and indexed with NEBNext® Multiplex Oligos for Illumina® (96 Index Primers; New England Biolabs, Ipswich, MA, USA). Fragmentation was carried out using incubation conditions recommended by the manufacturer for samples with a RIN ≥ 7 to produce RNA sizes ≥ 300 bp (94 °C for 10 min) with first strand synthesis carried out by incubation at 42 °C for 50 min. Modification of the manufacturer's conditions was used when enriching Adaptor-Ligated DNA for libraries with a size of 300 bp and 13 cycles of PCR were performed for final library enrichment. Resulting libraries were quantified using the Qubit 2.0 spectrophotometer (Life Technologies, California, USA) and average fragment size assessed, using the Agilent 2200

Tapestation (Agilent Technologies, Waldbronn, Germany). Sample libraries were combined in equimolar amounts into a single pool. The final library pool was sequenced twice on the NextSeq 500 at 1.1 pM and 75 bp paired-end reads were generated for each library using the Illumina NextSeq®500 v2.5 High-output 150 cycle kit (Illumina Inc., Cambridge, UK). A total of 319,920,579 reads (average of 13,330,024 ± 3,068,802 reads per sample) were generated for the 24 liver samples.

**Reduced representation bisulfite sequencing**. A total of 100 ng of total DNA was diluted and processed using the Premium RRBS Kit (Diagenode, Denville, NJ, USA) as per the manufacturer's instructions. Briefly, DNA was digested with MspI prior to end repair, adapter ligation and size selection. Products were then amplified by qPCR and pooled in equal amounts. Pooled libraries were bisulphite converted and PCR enriched following a second qPCR amplification. Libraries were quantified using the Qubit® 2.0 Fluorometer (Life Technologies) followed by quality assessment using the Agilent 2200 Tapestation and DS1000 screentape (Agilent Technologies). Pooled libraries were loaded at 1.1 M with 20% standard PhiX library (Illumina, CA, USA) and sequenced to 75 base pair single end on a NextSeq 500 (Illumina, CA, USA). Data were aligned to the rat reference genome Rn6 with Bismark[76]. A total of 407,904,185 reads (average of 16,996,008 ± 4,648,420 reads per sample) were generated for the 24 liver samples.

**Metabolomics**. Metabolomics analysis was conducted under contract with Metabolon Inc. (Durham, NC, USA) on four independent instrument platforms, as previously described[77]: two different separate reverse phase ultrahigh performance liquid chromatography-tandem mass spectroscopy analysis (RP/UPLC-MS/MS) with positive ion mode electrospray ionization (ESI), a RP/UPLC-MS/MS with negative ion mode ESI, as well as by hydrophilic interaction chromatography (HILIC)/UPLC-MS/MS with negative ion mode ESI.

All UPLC-MS/MS methods utilized a Waters ACQUITY UPLC and a Thermo Scientific Q-Exactive high-resolution/accurate mass spectrometer interfaced with a heated electrospray ionization (HESI-II) source and Orbitrap mass analyser operated at 35,000 mass resolution. The sample extract was dried and then reconstituted in solvents compatible to each of four methods used. Each reconstitution solvent contained a series of standards at fixed concentrations to ensure injection and chromatographic consistency. One aliquot was analysed using acidic positive ion conditions, chromatographically optimized for more hydrophilic compounds. In this method, the extract was gradient eluted from a C18 column (Waters UPLC BEH C18-2.1 × 100 mm, 1.7 μm) using water and methanol, containing 0.05% perfluoropentanoic acid (PFPA) and 0.1% formic acid (FA). Another aliquot was also analysed using acidic positive ion conditions, chromatographically optimised for more hydrophobic compounds. In this method, the extract was gradient eluted from the same afore-mentioned C18 column using methanol, acetonitrile, water, 0.05% PFPA and 0.01% FA and was operated at an overall higher organic content. Another aliquot was analysed using basic negative ion optimised conditions using a separate dedicated C18 column. The basic extracts were gradient eluted from the column using methanol and water, with 6.5 mM ammonium bicarbonate at pH 8. The fourth aliquot was analysed via negative ionization following elution from a HILIC column (Waters UPLC BEH Amide 2.1 × 150 mm, 1.7 μm) using a gradient consisting of water and acetonitrile with 10 mM ammonium formate, pH 10.8. The MS analysis alternated between MS and data-dependent MSn scans using dynamic exclusion. The scan range varied slightly between methods but covered 70–1000 $m/z$. Raw data were extracted, peak-identified and QC processed, using Metabolon's hardware and proprietary software. Peaks were quantified using area-under-the-curve.

**Shotgun metagenomics**. Samples of caecum content were collected at the time of sacrifice in two vials of 100 mg each and stored at −70 °C to perform evaluation of the gut microbiome. DNA was extracted from 100 mg caecum content using the Quick-DNA Faecal/Soil Microbe Miniprep Kit (Zymo Research, Irvine, CA, USA) with minor adaptations from manufacturer's instructions[78]. Adaptations were bead beating was performed at 5.5 m/s for three times 60 s using a Precellys 24 homogenizer (Bertin Instruments, Montigny-le-Bretonneux, France) and 2.50 μl elution buffer was used to elute the DNA, following which the eluate was run over the column once more to increase DNA yield. One negative extraction control (no sample added) and one positive extraction control (ZymoBIOMICS Microbial Community Standard; Zymo Research) were processed in parallel during the DNA extraction procedures and subsequently sequenced. DNA was quantified using the Qubit HS dsDNA Assay kit on a Qubit 4 fluorometer (Thermo Fisher Scientific, Horsham, UK).

Shotgun metagenomics was performed by GenomeScan (Leiden, The Netherlands). The NEBNext® Ultra II FS DNA module (cat# NEB #E7810S/L) and the NEBNext® Ultra II Ligation module (cat# NEB #E7595S/L) were used to process the samples. Fragmentation, A-tailing and ligation of sequencing adapters of the resulting product was performed, according to the procedure described in the NEBNext Ultra II FS DNA module and NEBNext Ultra II Ligation module Instruction Manual. The quality and yield after sample preparation were measured with the fragment analyzer. The size of the resulting product was consistent with the expected size of ~500–700 bp. Clustering and DNA sequencing using the NovaSeq6000 (Illumina inc.) was performed, according to manufacturer's

protocols. A concentration of 1.1 nM of DNA was used. NovaSeq control software NCS v1.6 was used.

The shotgun metagenomics data were pre-processed using the pre-processing package v0.2.2 (https://anaconda.org/fasnicar/preprocessing). In brief, this package concatenates reads, to remove Illumina adapters, discard low-quality (quality < 20 or >2 Ns) or too short reads (<75 bp), remove phiX and rat genome sequences, and finally sorts and splits the reads into R1, R2 and UN sets of reads.

Cleaned shotgun metagenomics reads were then processed for taxonomic and pathway profiling. Since there is no gold standard for computational analyses of shotgun metagenomics, we used a combination of approaches. We inferred the taxonomy with the RefSeq database on the metagenomics RAST server[79], IGGsearch (iggdb_v1.0.0_gut database)[80], MetaPhlAn version 2.9 (ref. [81]) and Kaiju 1.0.1 (ref. [82]).

**In vitro study of bacterial growth**. Azoxystrobin, boscalid, chlorpyrifos, imidacloprid and thiabendazole were diluted in dimethylsulfoxide (Merck, Feltham, UK) to obtain stock solutions. Glyphosate was diluted in water. Bacterial strains were provided by the Université de Caen Microbiologie Alimentaire (Caen, France) culture collections (Table S1). The broth dilution method was used to determine how pesticides modify bacterial growth under aerobic conditions. Bacteria from overnight broth cultures were re-suspended to $OD_{600 nm} = 0.3$ (*L. rhamnosus*) or $OD_{600 nm} = 0.2$ (*E. coli*) and further diluted 1000-fold in MRS without peptone for *L. rhamnosus*, or ABTG for *E. coli* broth, to obtain ~$10^5$ CFU/ml as confirmed in each experiment by plating cell suspension on agar plates incubated aerobically at 37 °C. Plates were incubated under aerobic conditions at 37 °C and inspected after 24 and 48 h. All experiments were performed in triplicate.

**DNA damage**. DNA damage was measured as the formation of AP sites in liver genomic DNA. AP sites are common DNA lesions caused by oxidative damage. We used the DNA Damage Assay Kit (AP sites, Colorimetric; Abcam, ab211154; Abcam plc, Cambridge, UK), according to manufacturer's instructions.

**Statistics and reproducibility**. The statistical analysis was performed using R version 4.0.0 (ref. [83]). Metabolome peak area values were median scaled, log transformed and any missing values imputed with sample set minimums, both on a per biochemical basis, and separately for each metabolome dataset. Statistical significance was determined using a Welch's two-sample $t$ test adjusted for multiple comparisons with FDR methods using the R package 'qvalue' version 2.17.0 (ref. [84]).

For the shotgun metagenomics, a compositional data analysis approach was used since gut metagenomics datasets are typically zero-inflated[85]. We used ALDEx version 2 (ALDEx2) for differential (relative) abundance analysis of proportional data[86]. Statistical analysis for taxa abundance was performed on a dataset corrected for asymmetry (uneven sequencing depths) using the *inter-quartile log-ratio* method, which identifies features with reproducible variance. A multivariate analysis consisting of a non-metric multidimensional scaling (NMDS) plot of Bray–Curtis distances between samples. Statistical significance of the sample clustering was evaluated with a PERMANOVA analysis on the Bray–Curtis distances with *adonis()* from vegan v2.4-2.

We also used OPLS-DA to evaluate the predictive ability of each omics approach. OPLS-DA is an extension of PLS methods, which includes an orthogonal component distinguishing the variability corresponding to the experimental perturbation (here the effects of the pesticide mixture) from the portion of the data that is orthogonal (that is, independent from the experimental perturbation. The R package ropls version 1.20.0 was used[87]. This algorithm uses the non-linear iterative partial least squares algorithm. Prior to analysis, experimental variables were centred and unit-variance scaled. Since PLS-DA methods are prone to overfitting, we assessed the significance of our classification using permutation tests (permuted 1000 times).

RNA-seq data were analysed with Salmon[88]. This tool was used to quantify transcript abundance by mapping the reads against a reference transcriptome (Ensembl Release Rattus Norvegicus 6.0 cDNA fasta). Mapping rate was 82.0 ± 4.4% on a rat transcriptome index containing 31,196 targets. The Salmon output was then imported in R using the Bioconductor package tximport. We created a transcript database containing transcript counts, which was used to perform a differential gene expression analysis using DESeq2 (ref. [89]). We finally used goseq to perform a GO analysis accounting for transcript length biases[90]. We also compared our transcriptome findings to a list of gene expression signatures collected from various rat tissues after treatments with various drugs, using the drugMatrix toxicogenomics database[91] with EnrichR[92].

DNA methylation calls from RRBS data were extracted with Bismark[76]. The output from Bismark was then imported in R and analysed with Methylkit[93]. DNA methylation calls were annotated using RefSeq gene predictions for rats (rn6 release) with the package genomation[94]. Other annotations were retrieved using the genome-wide annotation for rat tool org.Rn.eg.dbR package version 3.8.2. Statistical analysis was performed with logistic regression models fitted per CpG using Methylkit functions. $P$ values were adjusted to $Q$ values using SLIM method[95].

Statistical analyses of in vitro tests on bacterial growth were performed using GraphPad Prism version 8.0.1 (GraphPad Software, Inc, CA, USA). Differences

between treatment groups at different concentrations and the negative control were investigated using Kruskal–Wallis one-way ANOVA with Dunn's multiple comparison post-test.

**Reporting summary**. Further information on research design is available in the Nature Research Reporting Summary linked to this article.

## Data availability

Metabolomics raw data are available in Metabolights, with the accession number MTBLS138. Shotgun metagenomics raw data are available from the National Center for Biotechnology Information (NCBI), with BioProject accession no. PRJNA609596. Liver methylation raw data from the RRBS analysis is available at GEO accession no. GSE157551. The raw data from the transcriptomics analysis is available at GEO accession no. GSE157426. All other data are available from the corresponding author (or other sources, as applicable) on reasonable request.

## Code availability

The code used to perform the statistical analysis was compiled as an R Markdown document and made available (Supplementary Data 5). All data are freely available from a public GitHub repository (https://github.com/mesnage/MixtureTox/).

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

## Acknowledgements

This work was funded by the Sustainable Food Alliance (USA), and in part by the Franciscan Health Foundation (USA) and the Sheepdrove Trust (UK), whose support is gratefully acknowledged.

## Author contributions

M.N.A. and R.M. conceived the study with M.N.A. coordinating the investigation. R.M. and M.N.A. led the drafting of the manuscript with contributions from all authors. R.M. performed the bioinformatics and statistical analyses, and undertook interpretation of the data. M.T. conducted the bacterial growth assay. D.M., L.F. and F.B. conducted the animal treatment phase of the study. M.I. performed the DNA damage assay. Q.R.D. and R.D.Z. conducted the DNA extractions. C.A. and J.M.P. supervised the bacterial growth assay. E.B. conducted the RBBS analysis. E.S. conducted the RNA-seq. C.A.M. coordinated the RBBS and the RNA-seq analysis, and performed the RBBS data pre-processing.

## Competing interests

R.M. has served as a consultant on glyphosate risk assessment issues as part of litigation in the US over glyphosate health effects. The other authors declare no competing interests.
