## [Peer Review File · Communications Biology]

Reviewers' comments:

Reviewer #1 (Remarks to the Author):

In their manuscript "Multi-omics phenotyping of the gut-liver axis allows health risk predictability from subchronic toxicity tests of a low-dose pesticide mixture in Sprague-Dawley rats", Mesnage et al. report on a multi-omics toxicology study, in which the authors assessed the effects of a low-dose pesticide mixture in Sprague-Dawley rats. The authors selected six pesticides that are commonly detected in food products and exposed the animals for 90 days to their mixture in drinking water, each pesticide dosed at its acceptable daily intake level. Complementing standard toxicology endpoints (e.g., weight, histopathology, and serum biochemistry), this study included diverse omics measurements (metabolomics of serum and caecum, metagenomics, and liver transcriptomics/epigenomics). Interestingly, while exposure to the pesticide mixture did not significantly affect the standard toxicology endpoints, the omics measurements demonstrate a high sensitivity – especially, showing differentially abundant metabolites and liver transcripts. While a large number of molecules were significantly affected by pesticide exposure, the authors, especially, highlight effects on tryptophan and nicotinamide metabolism, suggesting that these reflect activation of a "cell danger response". Finally, the authors propose that the identified metabolites might serve as predictive biomarkers.

Mesnage et al.'s study is timely and clearly addresses an important topic. While we got a lot of data on the effects of individual compounds, it now becomes more and more important to assess how complex chemical mixtures that might be taken up over a day affect the homeostasis of our body and its environment. Systems toxicology approaches that leverage multiple sensitive molecular measurements can support the identification of toxicological mechanisms. For this, Mesnage et al.'s work provide a good and relevant example. Especially, when endpoints become more complex, detailed reporting on the used methods is critical and Mesnage et al. did a very good job, providing sufficient experimental details and sharing the data (only sharing of the computational scripts/code should be improved and availability of the generated data matrices would be helpful).

With this, Mesnage et al. work represents a high quality and scientifically relevant manuscript. However, before considering their manuscript for publication, I would like to ask the authors to address the following comments, especially to ensure that their conclusions accurately reflect the data:

- 1) The authors draw a couple of conclusions and/or propose possible links that appear fairly speculative given the data. I would suggest to revise and/or cut such speculations, if appropriate, e.g.:
 - The title claims that their approach "allows health risk predictability". Whether the observed effects predict any health risks has not been demonstrated and is doubtful – also considering the discussion by the authors that the observed effects might be adaptive rather than adverse in nature.
 - While molecular changes were detected, it is unclear how these observations could lead to validated biomarkers, esp. given that the observed changes do not appear too specific to the type of exposure and the mechanistic underpinnings are not fully clear. I would suggest not to emphasize the biomarker aspect too much and rather save this aspect for a dedicated biomarker discovery and validation study.
 - The author's argumentation toward a "cell danger response" is not clear to me (e.g., lines 605 – 620). This appears pretty speculative also given the subsequent discussion on an adaptive rather than adverse effect. Also, the support for an inflammatory response is absent and the support for an oxidative stress response is limited. Furthermore, the authors demonstrate similarity between the transcriptome response to pesticide and nicotinamide exposure (based on drugMatix), which speaks against a "cell danger response".
 - Lines 549 – 550 state "which reflects a metabolic adaptation that may exceed cellular capacity for homeostasis, ultimately leading to disease": no evidence is provided that the cellular capacity for detoxification is exceeded and the classical endpoints rather suggest otherwise. Clearly, a dose-

response study to identify the point of departure would be very interesting and I would encourage the authors to consider such study in the future. However, given the available data, I would rather conclude that the observed changes are likely adaptive without clear disease association.

- Lines 622 – 633 speculate about a possible link to autism. We know that things are usually more complex and a change in a single metabolite doesn't make a complex disease. With this, given the absence of any evidence, I would suggest to approach this topic with caution and rather cut this discussion point.

- Effects on Tryptophan metabolism are highlighted by the authors but kynurenine is not mentioned, although it appears upregulated in caecum. I would suggest to include the effects on kynurenine levels in the discussion.

2) While confirmed in the Editorial Policy Checklist, the code availability statement is missing from the manuscript. I would suggest that the authors include the analysis code in the supplement of their manuscript.

3) In addition, I would suggest that the authors make all derived data matrices directly available in the supplement (e.g., the raw and normalized metabolite matrix from Metabolon and the RNAseq count matrix for liver). This will support the reuse of the data.

4) The manuscript would benefit from a more extensive discussion on the benefits of (multi-) omics systems biology approaches, putting the study more clearly into the context of other published literature leveraging such (multi-) omics approaches.

Reviewer #2 (Remarks to the Author):

The manuscript entitled: "Multi-omics phenotyping of the gut-liver axis allows health risk predictability from subchronic toxicity tests of a low-dose pesticide mixture in Sprague-Dawley rats" presents an interesting study regarding the use of metabolomics and metagenomics in predicting the toxicity of chemicals at the low dose, that classical approach failed to detect. It provides new proofs regarding the potentially detrimental effects of non-commercial chemical mixtures at doses considered safe by the authorities and how new methodologies can detect the early organ changes produced by exposure to the mixture of chemicals that is in line with doi: 10.1016/j.fct.2017.03.012.

The manuscript is overall well written, material and methods part provides enough information for the study to be reproduced and the results and conclusions are based on the results obtained.

I have the following suggestions to improve the manuscript:

1. In the material and methods part, please explain the use of only female rats and not both sexes.
2. When you specify the doses used for each pesticide that corresponds to their ADI doses please add also reference for these.
3. Did the tap water was controlled for impurities? Please specify in the manuscript.
4. Please specify what clinical parameters you detected weekly.
5. Please specify the method of scarification of the animals.
6. Other studies also showed changes in oxidative stress markers or inflammation profile after exposure to low doses of pesticides with different profiles between 6, 12 and 18 months of exposure that is in line with your findings doi: 10.1016/j.toxlet.2019.09.015; doi: 10.1016/j.toxlet.2019.04.005.

Response to Reviewers' Comments

Reviewer 1:

In their manuscript "Multi-omics phenotyping of the gut-liver axis allows health risk predictability from subchronic toxicity tests of a low-dose pesticide mixture in Sprague-Dawley rats", Mesnage et al. report on a multi-omics toxicology study, in which the authors assessed the effects of a low-dose pesticide mixture in Sprague-Dawley rats. The authors selected six pesticides that are commonly detected in food products and exposed the animals for 90 days to their mixture in drinking water, each pesticide dosed at its acceptable daily intake level. Complementing standard toxicology endpoints (e.g., weight, histopathology, and serum biochemistry), this study included diverse omics measurements (metabolomics of serum and caecum, metagenomics, and liver transcriptomics/epigenomics). Interestingly, while exposure to the pesticide mixture did not significantly affect the standard toxicology endpoints, the omics measurements demonstrate a high sensitivity – especially, showing differentially abundant metabolites and liver transcripts. While a large number of molecules were significantly affected by pesticide exposure, the authors, especially, highlight effects on tryptophan and nicotinamide metabolism, suggesting that these reflect activation of a "cell danger response". Finally, the authors propose that the identified metabolites might serve as predictive biomarkers.

Mesnage et al.'s study is timely and clearly addresses an important topic. While we got a lot of data on the effects of individual compounds, it now becomes more and more important to assess how complex chemical mixtures that might be taken up over a day affect the homeostasis of our body and its environment. Systems toxicology approaches that leverage multiple sensitive molecular measurements can support the identification of toxicological mechanisms. For this, Mesnage et al.'s work provide a good and relevant example. Especially, when endpoints become more complex, detailed reporting on the used methods is critical and Mesnage et al. did a very good job, providing sufficient experimental details and sharing the data (only sharing of the computational scripts/code should be improved and availability of the generated data matrices would be helpful).

With this, Mesnage et al. work represents a high quality and scientifically relevant manuscript. However, before considering their manuscript for publication, I would like to ask the authors to address the following comments, especially to ensure that their conclusions accurately reflect the data:

Our response. Thank you for your thorough review and for your positive opinion. We appreciate the time taken to carefully consider our submission. Your comments allowed us to improve the clarity of our manuscript and its reproducibility. We acknowledge that our manuscript contained speculative statements about potential long-term health effects. We have revised the whole manuscript.

1) The authors draw a couple of conclusions and/or propose possible links that appear fairly speculative given the data. I would suggest to revise and/or cut such speculations, if appropriate, e.g.:
- The title claims that their approach "allows health risk predictability". Whether the observed effects predict any health risks has not been demonstrated and is doubtful – also considering the discussion by the authors that the observed effects might be adaptive rather than adverse in nature.

Our response. This is true. The new title is 'Multi-omics phenotyping of the gut-liver axis reveals metabolic perturbations in Sprague-Dawley rats exposed to a low-dose pesticide mixture'.

- While molecular changes were detected, it is unclear how these observations could lead to validated biomarkers, esp. given that the observed changes do not appear too specific to the type of exposure and the mechanistic underpinnings are not fully clear. I would suggest not to emphasize the biomarker aspect too much and rather save this aspect for a dedicated biomarker discovery and validation study.

Our response. We have reformulated the sentences proposing the use of the observed changes as a biomarker.

- The author's argumentation toward a "cell danger response" is not clear to me (e.g., lines 605 – 620). This appears pretty speculative also given the subsequent discussion on an adaptive rather than adverse effect. Also, the support for an inflammatory response is absent and the support for an oxidative stress response is limited. Furthermore, the authors demonstrate similarity between the transcriptome response to pesticide and nicotinamide exposure (based on drugMatix), which speaks against a "cell danger response".

Our response. We have removed the interpretation that the changes are indicating a cell danger response to focus our comments on the adaptive changes to the situation of oxidative stress.

- Lines 549 – 550 state "which reflects a metabolic adaptation that may exceed cellular capacity for homeostasis, ultimately leading to disease": no evidence is provided that the cellular capacity for detoxification is exceeded and

the classical endpoints rather suggest otherwise. Clearly, a dose-response study to identify the point of departure would be very interesting and I would encourage the authors to consider such study in the future. However, given the available data, I would rather conclude that the observed changes are likely adaptive without clear disease association.

Our response. Thank you for this comment. We have added a few sentences in the discussion to suggest the realisation of studies including a dose-response analysis. We added that:

“A dose-response study to identify the point of departure for these effects could inform on whether metabolic perturbations exceed cellular capacity for homeostasis.”

- Lines 622 – 633 speculate about a possible link to autism. We know that things are usually more complex and a change in a single metabolite doesn't make a complex disease. With this, given the absence of any evidence, I would suggest to approach this topic with caution and rather cut this discussion point.

Our response. We agree and we have cut this discussion point.

- Effects on Tryptophan metabolism are highlighted by the authors but kynurenine is not mentioned, although it appears upregulated in caecum. I would suggest to include the effects on kynurenine levels in the discussion.

Our response. Changes in kynurenine levels were not reaching statistical significance, which prompted us to focus on other more significant changes in tryptophan metabolism.

2) While confirmed in the Editorial Policy Checklist, the code availability statement is missing from the manuscript. I would suggest that the authors include the analysis code in the supplement of their manuscript.

Our response. We are sorry for this omission. We added a paragraph with all relevant details:

“Data and code availability

Metabolomics raw data is available in Metabolights, with the accession number MTBLS138. Shotgun metagenomics raw data are available from the National Center for Biotechnology Information (NCBI), with BioProject accession no. PRJNA609596. Liver methylation raw data from the RRBS analysis is available at GEO accession no. GSE157551. The raw data from the transcriptomics analysis is available at GEO accession no. GSE157426. The code used to perform the statistical analysis was compiled as an R Markdown document and made available under Supplemental Material. All data is freely available from a public GitHub repository (<https://github.com/mesnage/MixtureTox/>).”

Please note that the repositories GSE157551 and GSE157426 contain data which is relevant for other manuscripts which are not published yet. The data will be made available upon publication of this manuscript. In the meantime, we created reviewers links if you would like to analyse this data.

The following secure token has been created to allow review of record GSE157551 while it remains in private status: `gjejaoaejfonfuz`

The following secure token has been created to allow review of record GSE157426 while it remains in private status: `gpapyokgrpifnwf`

3) In addition, I would suggest that the authors make all derived data matrices directly available in the supplement (e.g., the raw and normalized metabolite matrix from Metabolon and the RNAseq count matrix for liver). This will support the reuse of the data.

Our response. We agree. All the data matrices made available as supplementary data. They are stored in spreadsheets of 3 Excel Files which are described in the document ‘Supplemental Material’.

4) The manuscript would benefit from a more extensive discussion on the benefits of (multi-) omics systems biology approaches, putting the study more clearly into the context of other published literature leveraging such (multi-) omics approaches.

Our response. We have rewritten the discussion to broaden the discussion on the benefits of (multi-) omics systems biology approaches.

Among other changes made to address your earlier comments, we added:

“Omics methods first found a place in the field of toxicology when the US National Research Council published a vision and strategy statement on toxicity testing in 2007 calling for a transition toward high-throughput predictive and mechanistic chemical toxicity assessments⁶⁹. This is mostly reflected by the development of adverse outcome pathways (AOP), which are sequences of molecular and cellular events known to reflect the development of a pathological process⁷⁰. Some omics technologies such as transcriptomics can be used predict key events and outcomes in networks of AOPs⁷¹. Contemporary toxicology is increasingly using artificial intelligence models to predict toxicological properties⁷². These models can be based on pathway perturbations identified using omics data as suggested by a recent study showing that the training of a machine learning algorithm on the transcriptome changes caused by different endocrine disruptors in fish ovaries adequately identified anti-androgenic properties of the herbicide linuron⁷³”

Reviewer #2 (Remarks to the Author):

The manuscript entitled: “Multi-omics phenotyping of the gut-liver axis allows health risk predictability from subchronic toxicity tests of a low-dose pesticide mixture in Sprague-Dawley rats” presents an interesting study regarding the use of metabolomics and metagenomics in predicting the toxicity of chemicals at the low dose, that classical approach failed to detect. It provides new proofs regarding the potentially detrimental effects of non-commercial chemical mixtures at doses considered safe by the authorities and how new methodologies can detect the early organ changes produced by exposure to the mixture of chemicals that is in line with doi: 10.1016/j.fct.2017.03.012.

The manuscript is overall well written, material and methods part provides enough information for the study to be reproduced and the results and conclusions are based on the results obtained.

Our response. Thank you for the time spent analysing our manuscript, and for your overall positive opinion of our manuscript and suggestions to strengthen our study.

I have the following suggestions to improve the manuscript:

1. In the material and methods part, please explain the use of only female rats and not both sexes.

Our response. We added in our manuscript why we have chosen females as follows:

“Female animals were chosen to make the results of this investigation comparable to our previous multi-omics study showing that the long-term exposure to a glyphosate-based herbicide was associated with the development of liver disease (Mesnage et al. 2015; Mesnage et al. 2017).”

We agree that it would have been informative to test both sexes because it can reveal sex dimorphism. We added this limitation in our discussion:

“Because the liver is a sex-dimorphic organ, with sex differences in liver metabolism and in the sensitivities to liver diseases, the results of this study in females cannot be extrapolated to males.”

2. When you specify the doses used for each pesticide that corresponds to their ADI doses please add also reference for these.

Our response. Thank you for spotting these missing references. They were added.

3. Did the tap water was controlled for impurities? Please specify in the manuscript.

Our response. Thank you for this comment. We haven't tested the contamination of tap water because it is generally known to be negligible since the regulation is very strict in the EU with maximum residues limits for a given pesticide in tap water of 0.1 µg/L up to a maximum of 0.5 µg/L for all pesticides that may be present. In the case of glyphosate, this is approximately 20,000 times lower than the concentration in the drinking water of pesticide-exposed rats in our study. It is thus extremely unlikely that a possible contamination of tap water is a confounding factor in the results we obtained in our experiment. However, contamination of diets can be much higher as pesticides are routinely found in some commercial rat diets (doi:10.1371/journal.pone.0128429). Hence, we undertook a comprehensive analysis of the feed used in this study.

We added in the manuscript: “The presence of pesticides was not measured in the tap water. The regulation in the EU has maximum residues limits for a given pesticide in tap water of 0.1 µg/L, and up to a maximum of 0.5 µg/L for all pesticides that may be present. Given these low legal tolerance limits, it was not deemed necessary to analyse tap water for pesticide contamination since if present would be extremely low and unlikely to have any physiological consequences of the experimental animals.”.

4. Please specify what clinical parameters you detected weekly.

Our response. We clarified that *'Body weight, water and food consumption of experimental animals were measured before the start of the treatment and then weekly for 90 days.'*

5. Please specify the method of scarification of the animals.

Our response. We added that *'Each animal was anesthetized by inhalation of a mixture of 70% CO2 and 30% O2 and sacrificed by exsanguination from then vena cava after blood withdrawal for sample collection.'*

6. Other studies also showed changes in oxidative stress markers or inflammation profile after exposure to low doses of pesticides with different profiles between 6, 12 and 18 months of exposure that is in line with your findings doi: 10.1016/j.toxlet.2019.09.015; doi: 10.1016/j.toxlet.2019.04.005.

Our response. This is very true. Thank you for the suggestion. We have added this remark towards the end of the discussion and cited the two suggested studies:

"In support of this possibility are findings from other studies, which showed changes in oxidative stress markers or inflammation profiles after exposure to low doses of pesticides with different profiles between 6, 12 and 18 months of administration (Docea et al., 2019 ; Fountoucidou et al., 2019)"

REVIEWERS' COMMENTS:

Reviewer #1 (Remarks to the Author):

Thanks for addressing my previous comments. I recommend the current version of the manuscript for publication.

Reviewer #2 (Remarks to the Author):

well-justified revision
manuscript accepted